# Predicting the Potential Distribution of *Oxalis debilis* Kunth, an Invasive Species in China with a Maximum Entropy Model

**DOI:** 10.3390/plants12233999

**Published:** 2023-11-28

**Authors:** Xinsheng Qin, Mingxin Li

**Affiliations:** College of Forestry and Landscape Architecture, South China Agricultural University, Guangzhou 510642, China; mingx.lee@foxmail.com

**Keywords:** potential geographic area, MaxEnt model, environmental variable, *Oxalis debilis*, invasive plant

## Abstract

*Oxalis debilis* Kunth, an invasive plant native to South America, has already spread extensively throughout various regions in China including West China, East China, Central China, and South China. It poses a certain degree of damage to the local ecosystem and demonstrates significant invasive potential. Utilizing distribution information along with environmental variables such as bioclimate, soil factors, elevation, and UV-B radiation, the MaxEnt model combined with ArcGIS was employed to forecast the potential distribution of *O. debilis* in China. The ROC curve was employed to assess the accuracy of the model, while the jackknife test was utilized to identify dominant environmental variables and determine their optimal values. The simulated AUC value was 0.946 ± 0.004, and the predicted results exhibited a remarkable concordance with the actual outcomes, thereby indicating that the Maxent model demonstrated a high level of confidence in its predictive capabilities. The potential distribution of *O. debilis* in China spanned 18,914,237 km^2^, accounting for 19.70% of the total land area. This distribution was primarily observed in East, Central, and South China, with Guangdong, Guangxi, and Guizhou being identified as highly suitable habitats for *O. debilis*. Furthermore, it was observed that the distribution of *O. debilis* is primarily influenced by environmental variables such as the precipitation of the driest month, the monthly diurnal range, the mean temperature of the wettest quarter, and the isothermality. The findings can serve as a valuable point of reference for the prevention and monitoring of *O. debilis* spread, thereby contributing to the protection of China’s agricultural, forestry, and ecological environments. It is imperative to acknowledge the hazards associated with *O. debilis*, closely monitor its invasion, and prevent uncontrolled dissemination.

## 1. Introduction

*Oxalis debilis* Kunth is a significant perennial herb with medicinal and edible properties as well as environmental uses, commonly referred to as pink wood sorrel, which belongs to the family Oxalidaceae [1]. The plant species is indigenous to Central and South America and has a global distribution, including India, particularly in the north-eastern states [2]. It is a tuberous geophyte and primarily thrives in the subtropical biome. *O. debilis* was introduced to China as an ornamental plant in the 19th century [3]. The species is currently widely distributed throughout East, Central, and South China, with notable populations in Guangxi, Hebei, Fujian, Shaanxi, Sichuan, and Yunnan [4]. The species can be readily observed in low mountain ranges, rice fields, agricultural lands, horticultural areas, road verges, residential compounds, public parks, and lawns [5]. Due to its high ornamental value in terms of color and leaf shape, it is extensively utilized for landscaping purposes [6] (Figure 1).

*O. debilis* thrives in a sunny, warm, and humid environment. It is tolerant to barren conditions, acid–alkaline soil, and a drought climate. The plant possesses a wide range of adaptability, strong vitality, and cold resistance [7], enabling rapid growth even during the monsoon season [8]. In recent years, the population of *O. debilis* has undergone rapid growth, leading to an expanded distribution range. Consequently, the ecological impact of *O. debilis* on local environments and surrounding fauna has become increasingly evident, necessitating heightened attention and vigilance from individuals. First, as an alien species, it has the potential to cause ecological disasters. This plant primarily propagates through subterranean bulbs, rendering them highly invasive [9]. Additionally, it exhibits strong allelopathic effects that can impede the growth of other crops and weeds, thereby exerting a significant impact on the ecosystem structure and even posing a threat [10]. Second, it will engender substantial economic ramifications. As an outbreak weed, it has a serious impact on the growth of farmland crops and garden greening plants; for example, the yield of potatoes, peppers, and watermelons grown near *O. debilis* have all experienced various degrees of reduction of up to a 100% reduction [11], which will lead to serious economic losses. According to the risk analysis methods of pests at home and abroad, it was found that the risk value of *O. debilis* was 1.94, indicating that it is a medium or even high-risk plant [12,13].

In recent years, numerous methods have been proposed to predict the species potential distribution using niche models, including BIOCLIM, GARP, CLIMEX, Domain, MaxEnt [14,15]. These models share a similar underlying principle and are all based on known geographic distribution data of a species along with corresponding environmental variables. Specific algorithms are used to calculate the ecological requirements of the species [16], and then project the operation results into a different space and time to predict the potential distribution of the species [17,18]. As a result, these models are extensively employed for the assessment of invasion risks posed by foreign pests, conservation of endangered flora and fauna, as well as analysis of deforestation processes and ecological degradation [19]. The maximum entropy method primarily aims to determine the maximum entropy of known distribution points for species and environmental variables, enabling the prediction and evaluation of species distribution [20]. This approach is characterized by its requirement for a small sample size and minimal sample bias [21]. The economic, social, ecological, and other impacts of *O. debilis* as an invasive plant are evident. This research used MaxEnt to predict the potential distribution of *O. debilis* in China based on the occurrence records of *O. debilis* in China and comprehensively analyzed the key environmental variables affecting its distribution. The objective of this research was to provide an important reference and a theoretical foundation for monitoring, prevention, control, and resource utilization of *O. debilis.*

## 2. Results

### 2.1. Verification of Model Accuracy

The simulation results of this study demonstrated that the MaxEnt model achieves an average AUC value of 0.946 (standard deviation ± 0.004) on the training data after 10 repeated runs, indicating its strong predictive performance (Figure 2).

### 2.2. Predicting Habitat Suitability of O. debilis in China with MaxEnt

The prediction results indicated that the suitable areas of *O. debilis* in China were widespread, mainly centered in the east and south while a few areas in the western and northeastern regions were poorly suited for it (Figure 3).

The potential suitable area of *O. debilis* in China was approximately 1,891,4237 km², representing 19.70% of the country’s total land area. Within the area, highly suitable habitat encompassed 563,700 km², accounting for roughly 5.87% of the land area and mainly concentrated in the southern region and coastal area. The first part comprised Eastern China which exhibited a wide distribution but was particularly concentrated in Jiangsu, Shanghai, Zhejiang, Central Jiangxi, as well as Taiwan and Fujian coastal regions. The second part was located in Central China, primarily concentrated in Hunan. The third part covered the eastern portion of the southwest regions, mainly distributed in the junction of Sichuan, Chongqing, and the east of Guizhou. Most of South China, particularly Guangxi and Guangdong, was covered by the fourth part. These findings suggested a higher likelihood of reproductive success for *O. debilis* in these areas, indicating an elevated risk of invasion.

The moderately suitable habitat covered 600.75 million km², or 6.33% of the total area, and was mainly in Anhui, Jiangxi, Hubei, Hunan, Sichuan, Chongqing, and Guizhou, with scattered occurrences in the south-eastern coastal areas, South China, Yunnan, and Tibet, suggesting that the introduction and invasion of *O. debilis* in these areas poses a moderate risk.

The poorly suitable habitat covered an area of 720,200 km², which represents 7.50% of the total land area. The plant had a wide distribution range, mainly in Southeast China, with limited distribution in central Henan and Shaanxi, as well as Western Yunnan, Tibet, and Xinjiang. These findings suggested that *O. debilis* had a low potential to colonize these areas.

### 2.3. Environmental Variables That Affect the Prediction of Suitability

SpH (Soil pH), UV-B (Annual Mean UV-B), Bio2 (Monthly diurnal range), Bio3 (Isothermality), Bio8 (Mean Temperature of wettest quarter), Bio10 (Mean temperature of the warmest quarter), Bio14 (Precipitation of the driest month), Bio15 (Precipitation seasonality), Bio17 (Precipitation of the driest quarter), and Elve (Elevation) were obtained after checking for collinearity. The MaxEnt model was applied to assess the relative importance of each environmental variable by analyzing their contribution rates to the potential distribution area of *O. debilis*. The correlation coefficients between environmental variables are presented in Table 1.

The results showed that Bio14 was the most important environmental variable affecting the distribution of *O. debilis* (Table 2), with a contribution rate of up to 72.1%. It was followed by Bio2, Bio8, Bio3, and UV-B, which contributed 5.9%, 5.0%, 3.8%, and 3.5%, respectively. The environmental variables with little influence were Elev, Bio10, Bio15, SpH, and Bio17, with a total contribution rate of 9.8%. Briefly, the main environmental variables influencing the potential distribution of *O. debilis* were Bio14, Bio2, Bio8, and Bio3. It is noteworthy that Bio14 exhibits the highest contribution rate, reaching an impressive 72.1%.

The jackknife test was used to determine the importance of the environmental variables for *O. debilis*. The jackknife test results (Figure 4) suggested that the environmental variable had the greatest influence on the prediction gain of the model training when a single environmental variable acts. The gain values of Bio14, Bio2, and Bio17 were all greater than 1.3 when acting alone, showing that water and temperature were the main factors affecting the prediction results of suitability of *O. debilis*. Meanwhile, Bio10 and Bio15 also had a large influence on the gain of model training and prediction, as their mean values fluctuated around 1.0. Bio3 had minor impact on the gaining of prediction results when it acted alone.

### 2.4. Effects of Environmental Variables on the Habitat Adaptability of O. debilis

By analyzing the response curves of four major environmental variables affecting the potential distribution of *O. debilis* (Figure 5), we can determine the specific temperature and precipitation ranges that affect the growth of *O. debilis*. When the precipitation of the driest month (Bio14) dropped below 20 mm, the habitat adaptability of *O. debilis* decreased to less than 0.3, whereas an increase in precipitation significantly enhanced its habitat adaptability. Conversely, when the precipitation in the driest month exceeded 60 mm, there was a consistent habitat adaptability for *O. debilis* at approximately 1.0 (Figure 5a). The habitat adaptability of *O. debilis* was deemed unsuitable when the monthly diurnal range (Bio2) fell below 5 °C or exceeded 14 °C, exhibiting an adaptability lower than 0.2. As the monthly diurnal range increased, the habitat adaptability demonstrated a sharp rise and reached its peak at approximately 6 °C, reaching nearly 0.84; however, beyond this threshold, the adaptability gradually declined and dropped below 0.5 when the temperature difference surpassed 9 °C (Figure 5b). The habitat adaptability of *O. debilis* initially increased with the rise in mean temperature during the wettest quarter (Bio8). When the mean temperature during the wettest quarter was below 15 °C, the habitat adaptability of *O. debilis* remained below 0.2, reaching a peak value of 0.75 at 26 °C. Subsequently, there was a decline in habitat adaptability. Once the mean temperature during the wettest quarter exceeded 32 °C, the habitat adaptability stabilized at 0.35. This indicates that *O. debilis* exhibits a preference for warm and humid environments (Figure 5c). When the isothermality (Bio3) ranged from 15 to 20, it stabilized at 0.8 and increased with an elevated ratio of diurnal temperature difference to annual temperature difference, reaching a peak of 0.9 at 23. Subsequently, the habitat adaptability gradually declined and dropped below 0.2 when the isothermality exceeded 45. However, as long as the isothermality was maintained within the range of 15–38, the adaptability consistently remained above 0.5 (Figure 5d). The factors influencing the diurnal and annual temperature ranges included latitude, terrain characteristics, topography, underlying surface properties, land-sea distribution, and vegetation status. Diurnal temperature difference can be increased by low latitudes, valleys with lower elevation, and land areas with higher terrain features, while annual temperature difference can be increased by higher latitudes, low-lying lands, and areas with less vegetation cover. Therefore, it can be concluded that environments characterized by medium to low latitudes, coastal regions with flat terrains and certain levels of vegetation cover are more favorable for the growth of *O. debilis*.

In conclusion, *O. debilis* is more likely to be found in a warm and humid environment characterized by minimal temperature fluctuations between day and night as well as throughout the year. The precipitation during the driest month exceeded 40 mm, while the monthly diurnal range ranged from 6 °C to 8 °C. Moreover, the mean temperature of the wettest quarter fell within the range of 22 °C to 30 °C, and isothermality values varied between 15 and 24, making it an optimal habitat for survival.

## 3. Discussion

### 3.1. The Predictive Accuracy and Contemporary Issues of the MaxEnt Model

The MaxEnt model integrates species distribution data with corresponding environmental variables to find the maximum entropy of species distribution patterns, enabling the accurate prediction of potential species distributions. This method offers high precision and significant practical implications [22,23]. Additionally, the MaxEnt model exhibits a strong reliance on the species distribution data. In general, a larger dataset of distribution points improves the accuracy and reliability of prediction results [24]. In this research, a substantial number of distribution points were selected, and the prediction results were evaluated by accuracy using ROC curves. The results demonstrate that this model exhibits excellent predictive performance and high reliability for predicting the distribution area of *O. debilis*. However, the MaxEnt model run in this research is based on the assumption of ideal ecological conditions and does not consider the influence of other factors like soil type, interspecific relationships, species adaptations, geology, topography, etc., which are all factors that need to be taken into account [25]. Consequently, there could be a potential discrepancy between the predicted outcomes and the actual distribution range of *O. debilis*. Moreover, it is imperative to incorporate additional variables beyond environmental factors in order to augment the predictive accuracy of the model, thus necessitating further investigation.

### 3.2. The Distribution Characteristics of O. debilis and Its Response to Environmental Factors in the Current Period

Among the biological environmental factors, climatic variables (temperature and precipitation) play a pivotal role in determining the potential geographic distribution of species [26]. This research revealed that within the contemporary climatic context, the suitable habitats of *O. debilis* were predominantly distributed across central, eastern, and southern regions of China, with a concentration of medium to high suitability observed in Guangxi, Guangdong, Guizhou, and Yunnan. The model demonstrated that *O. debilis* exhibited the highest sensitivity to precipitation of the driest month, and its fitness level increased significantly with precipitation, reaching its peak when precipitation of the driest month exceeded 60 mm. The results of the model simulation are consistent with the main distribution areas of *O. debilis*, which are characterized by being relatively humid in the dry season [27]. It demonstrates that *O. debilis* exhibits a high degree of sensitivity to variations in precipitation. The maximum habitat adaptability value of 0.75 was observed at a mean temperature of 26 °C during the wettest quarter, indicating a preference for warm and humid climatic conditions of this species. The leaves will be scorched by excessively high temperatures, while insufficiently low temperatures will hinder blooming. This aligns closely with the habitat description of *O. debilis* community in Zhou et al. [28].

### 3.3. Comprehensive Treatment of O. debilis

*O. debilis* has a large seed quantity, easy separation of bulbs, easy propagation of stems along with soil seedlings, rapid asexual reproduction, strong adaptability [29], fast growth rate, strong regeneration performance, high reproduction rate, strong encroachment, and extremely strong stress resistance. It inflicts significant harm to the low-growing herbs in its vicinity [30] and is exceptionally challenging to eradicate. Meanwhile, the shade-tolerant evergreen ground cover, *O. debilis*, exhibits a well-organized growth pattern with densely packed leaves, vibrant flowers, and an extended period of blooming [31]. Due to its high aesthetic value, many individuals perceive it as an ornamental plant suitable for cultivation or unrestricted harvesting, thereby facilitating its potential spread and consequently elevating the associated risks. Therefore, enhanced publicity and education should be conducted to increase public awareness of prevention measures. This involves establishing and enhancing the database for invasive alien organisms and plant pests, improving the data quality and functionality of existing domestic databases, and optimizing their practicality and user-friendliness, thereby creating favorable conditions for promoting scientific knowledge dissemination, engaging public participation, and fostering the advancement of citizen science [32]. Relevant quarantine agencies need to be established while strictly implementing quarantine protocols to effectively control species introductions. The concerned regulatory bodies should strengthen monitoring efforts for early detection and prevention purposes. Rational control measures must be implemented to impede the further spread of *O. debilis*. The integration of remote sensing technology and mathematical models through comprehensive research will enable a more precise understanding of the ecological risks faced by species, facilitating the identification of effective solutions.

## 4. Materials and Methods

The research methodology is illustrated in Figure 6. First, data collection and pre-processing were conducted. Second, the MaxEnt model was executed to predict potential distribution. Third, a comprehensive analysis and evaluation of the obtained data were performed.

### 4.1. Data Sources

#### 4.1.1. Software Sources

The ArcGIS software (Version 10.7; http://www.arcgis.com/features/index.html (accessed on 20 May 2020)) was used for data pre-processing for MaxEnt and image rendering. The MaxEnt software (Version 3.4.3; https://biodiversityinformatics.amnh.org/open_source/maxent/ (accessed on 7 July 2023)) was used to predict the potential distribution range of this species, and to obtain the ROC curve, percentage contributions of environmental variables, jackknife test, and response curves. The SPSS (http://www.spss.com.cn (accessed on 14 July 2023)) was used to examine the correlation, while the highly correlated environmental variables were removed.

#### 4.1.2. Occurrence Records of *O. debilis* in China

The data sources for *O. debilis* distribution points were: 1. Specimen records: the Chinese Virtual Herbarium (CVH, http://www.cvh.ac.cn/ (accessed on 26 June 2023)), the National Specimen Information Infrastructure (NSII, http://www.nsii.org.cn/ (accessed on 28 June 2023)), the Plant Photo Bank of China (PPBC, http://ppbc.iplant.cn/ (accessed on 29 June 2023)), and the Global Biodiversity Information Facility (GBIF, https://www.gbif.org/ (accessed on 29 June 2023)). Record the address indicated in the image based on the query results. 2. Literature reports: Search the published journals and academic papers on *O. debilis* in China to obtain its geographic distribution information [33,34,35,36]. In cases where records lacked precise geographic coordinates, Amap maps (https://ditu.amap.com/ (accessed on 30 June 2023)) were employed to ascertain latitude and longitude values accurately, as well as to eliminate any duplicate distribution points. Among them, due to the absence of precise geographical coordinates, certain data are assigned to the administrative unit center at the most granular level of specimen markers. To comply with the MaxEnt software requirements, the distribution points of *O. debilis* were arranged in the order of species name, longitude, and latitude to generate CSV format files. All obtained distribution point data underwent a process to remove duplicate items and redundant points. This process retained one distribution point every 2.5 arc-minutes, resulting in a total of 299 valid data points. These data points were saved in CSV format for ArcGIS to perform the subsequent analysis [37] (Figure 7).

#### 4.1.3. Selection of Environmental Variables

Climate, landform, and soil are crucial environmental factors that affect plant growth and distribution [38]. In this research, 19 bioclimatic factors, 1 altitude factor, 2 soil factors, and the UV-B4 radiation factor were selected as initial environmental variables. These included temperature related (Bio01–Bio11) and precipitation related (Bio12–Bio19), 19 biological climate factors, and one global altitude factor (Elev) with 2.5 min spatial resolution obtained from the World Climate Database (http://www.worldclim.org/ (accessed on 14 July 2023)) [39]; soil pH (SpH) and soil organic carbon (SC) from the Center for Sustainability and the Global Environment (http://www.sage.wisc.edu/atlas/index.php (accessed on 14 July 2023)) [40]; and global UV-B radiation from the gIUV database (http://www.ufz.de/gluv/ (accessed on 14 July 2023)) [41] (Table 3).

First, ArcGIS was used to unify the spatial reference of these 23 variables into the geographic coordinate system GCS_WGS_1984. The tool also converted all layers into ASCII format and unified the pixel and range of each layer before converting them all into TIF format. Second, these were overlaid onto a 1:4,000,000 scale base map and an administrative division map of China, obtained from the National Geomatics Center of China (http://nfgis.nsdi.gov.cn/ (accessed on 14 July 2023)), to extract the environmental data [41].

### 4.2. Methods

#### 4.2.1. Testing and Screening for Correlation of Environmental Variables

Not all environmental variables were found to impact the prediction of *O. debilis* distribution when using the MaxEnt model. As a result, it was necessary to exclude the environmental variables with a low contribution rate and retain variables with a contribution rate greater than 1.0% for model fitting to make accurate predictions [42,43]. The Extract Multi Value to Points function of the Spatial Analyst Tool of ArcGIS was used to extract the environmental variable values corresponding to 299 distribution points. The Pearson correlation coefficient method was then employed in SPSS to calculate the correlation between the remaining environmental variables. If the absolute value of the correlation coefficient exceeded 0.8, it indicated collinearity between the variables, and only one environmental factor needed to be retained [44,45].

#### 4.2.2. MaxEnt Modeling Process

The latitude and longitude data of *O. debilis* in China and screened environmental variables were imported into MaxEnt software to predict the potential geographical distribution of *O. debilis* in China in the context of modern climate change. The specific parameters of the model were set as follows: the random test percentage was set to 25%, 25% of the samples were used as the test set, and the remaining 75% as the training set. The replicated run type was chosen as bootstrap, with the number of replicates set to ten times. The prediction result was computed as the average value of the results obtained from the ten operations. The options ‘Create response curves’, ‘Make pictures of predictions’, and ‘Do jackknife to measure variable importance’ were selected. The output format was set to ‘Cloglog’ and the output type was set to ‘ASC’. Other basic settings were set to their default values.

#### 4.2.3. Maxent Model Accuracy Checking

The ROC curve is widely used in the evaluation of the species potential distribution prediction model which is a highly recognized diagnostic test evaluation index. The area under the ROC curve (AUC value), ranging from 0 to 1, represents the model’s performance, with values closer to 1 indicating superior performance [46,47].

#### 4.2.4. Categorization of the Suitable Habitat Ranks

The MaxEnt output was imported into ArcGIS, converted to raster data by Conversion tools in the toolbar, and then the Reclassify program was used in Spatial analyst tools, and Natural breaks (Jenks) selected to divide the four classes of existence probability (P) of *O. debilis*, ‘unsuitable habitat’ (0 < P ≤ 0.1), ‘poorly suitable habitat’ (0.1 < P ≤ 0.3), ‘moderately suitable habitat’ (0.3 < P ≤ 0.5), and ‘highly suitable habitat’ (0.5 < P ≤ 1) [12].

## 5. Conclusions

The management of invasive alien species and their influence is a vital ecological protection challenge in China. Using ArcGIS techniques and the MaxEnt model, we accurately predicted areas conducive to the growth of *O. debilis*. The research revealed that the ideal environment for *O. debilis* growth was found mainly in East China and Central China, covering an estimated 563,700 km^2^. These habitats offer a consistent, balmy, and humid climate. The crucial environmental variables impacting the distribution of this species are the precipitation of the driest month, the monthly diurnal range, the mean temperature of the wettest quarter, and the isothermality. The findings of this research could enable environmental managers to promptly implement measures to prevent further spread of this species in mainland China.

## Figures and Tables

**Figure 1 plants-12-03999-f001:**
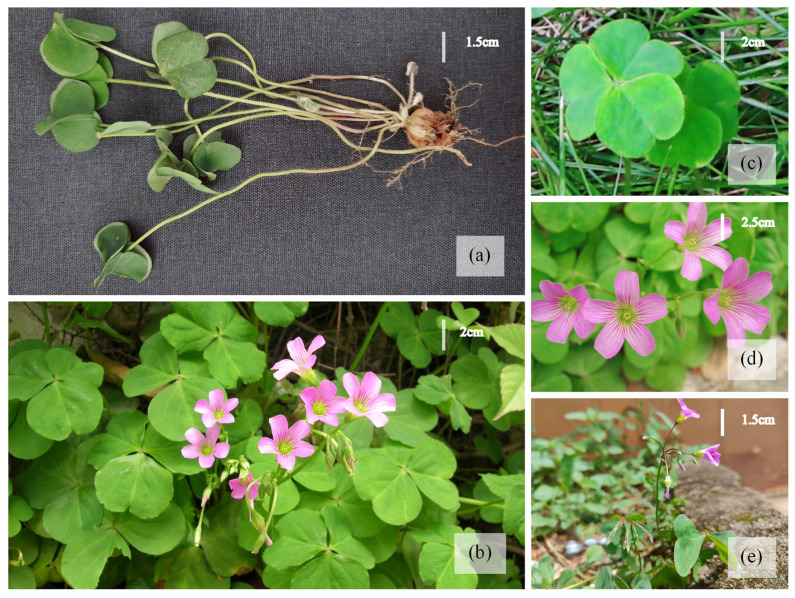
Photograph of *Oxalis debilis* Kunth. (**a**) A whole plant with a shoot and root system; (**b**) growing in the lawn; (**c**) leaves; (**d**) flowers; (**e**) plant in a flower bed.

**Figure 2 plants-12-03999-f002:**
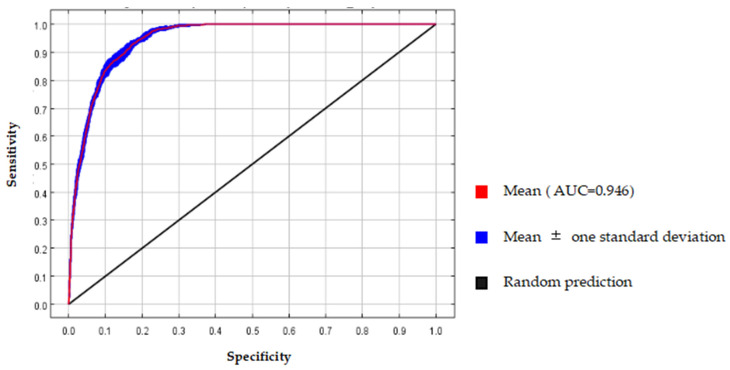
Operating characteristic curve of *Oxalis debilis* Kunth predicted by MaxEnt model.

**Figure 3 plants-12-03999-f003:**
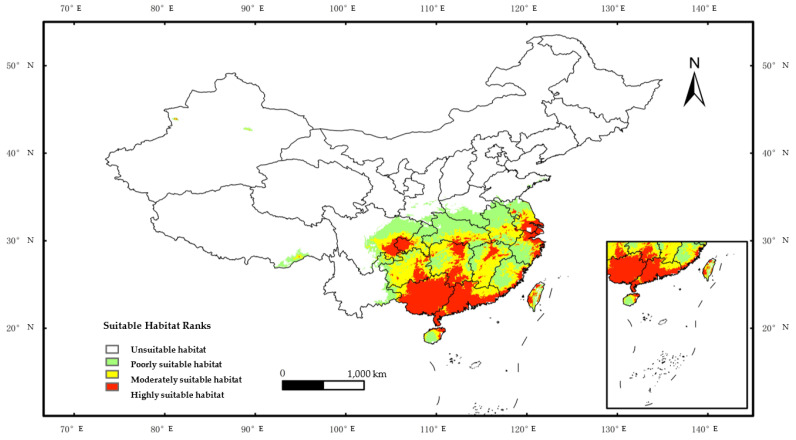
The potential current habitat suitability of *Oxalis debilis* Kunth according to occurrence records in China.

**Figure 4 plants-12-03999-f004:**
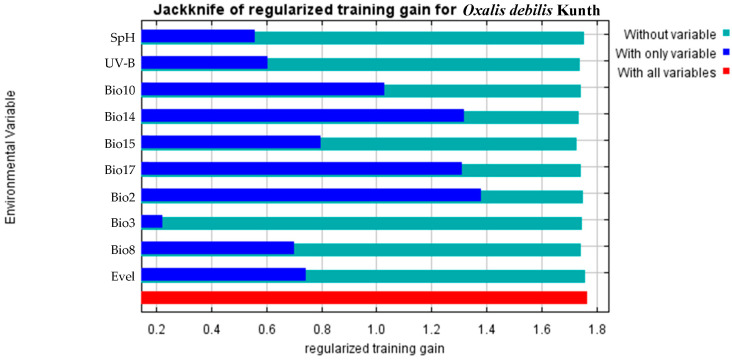
The jackknife of the regularized training gain for environmental variables.

**Figure 5 plants-12-03999-f005:**
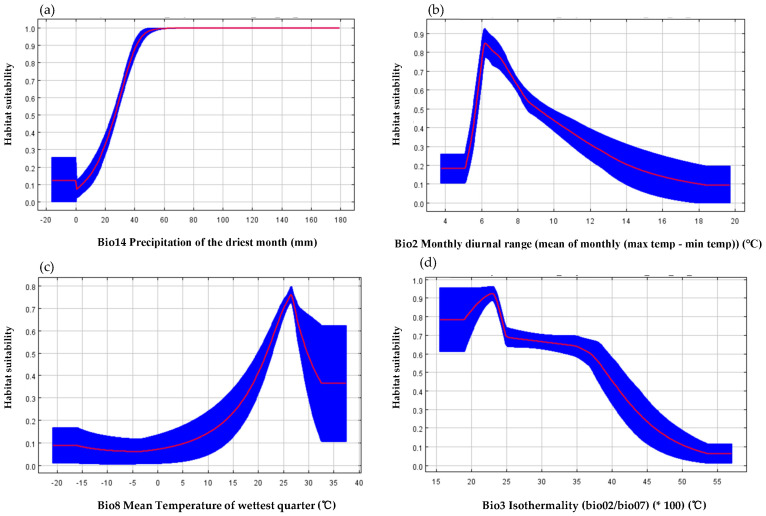
Response curve using only the corresponding variable. (**a**) Bio14 Precipitation of the driest month; (**b**) Bio2 Monthly diurnal range (mean of monthly (max temp − min temp)); (**c**) Bio8 Mean Temperature of wettest quarter; (**d**) Bio3 Isothermality (bio02/bio07) (×100).

**Figure 6 plants-12-03999-f006:**
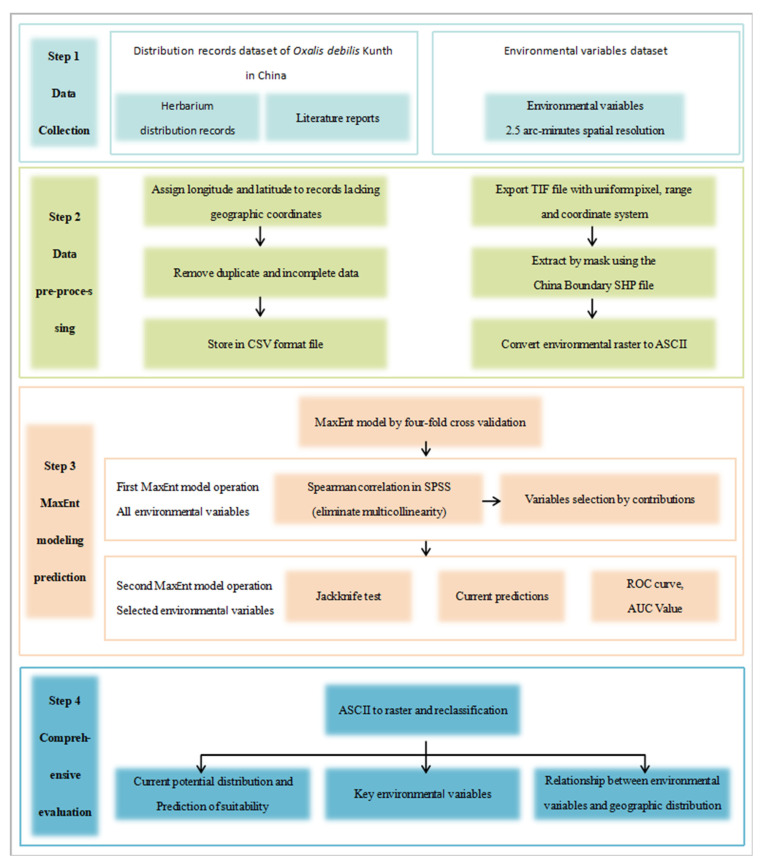
The framework of the operation process.

**Figure 7 plants-12-03999-f007:**
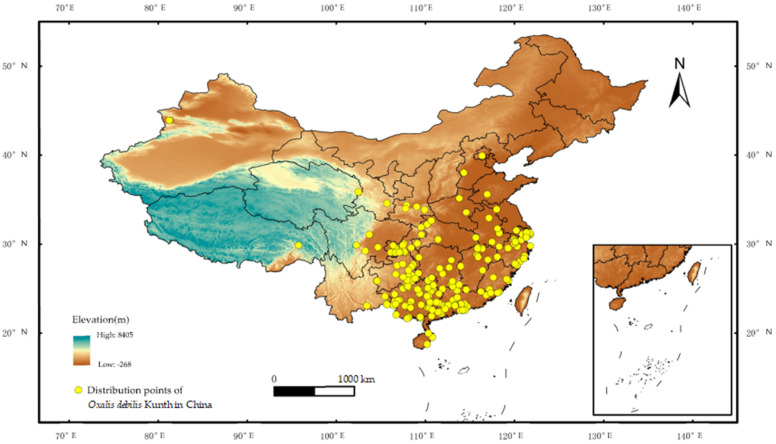
Distribution points of *Oxalis debilis* Kunth in China.

**Table 1 plants-12-03999-t001:** Pearson correlation coefficients among the important environmental variables.

Variables	SpH	UV-B	Bio2	Bio3	Bio8	Bio10	Bio14	Bio15	Bio17
UV-B	−0.431								
Bio2	0.211	−0.256							
Bio3	−0.344	0.758	0.275						
Bio8	0.128	0.334	−0.329	0.165					
Bio10	−0.095	0.314	−0.294	0.006	0.705				
Bio14	−0.153	0.075	−0.446	−0.405	0.043	0.437			
Bio15	−0.173	0.329	0.244	−0.630	0.150	−0.126	−0.635		
Bio17	−0.247	0.096	−0.397	−0.353	−0.054	0.427	0.966	−0.585	
Elve	−0.170	−0.054	0.269	0.253	−0.682	−0.919	0.468	0.302	−0.422

**Table 2 plants-12-03999-t002:** Percentage contribution and permutation importance of the environmental variables that contribute more than 1%.

Variable	Description (Unit)	Percent Contribution (%)	Permutation Importance (%)
Bio14	Precipitation of the driest month (mm)	72.1	26.3
Bio2	Monthly diurnal range (mean of monthly (max temp − min temp)) (°C)	5.9	7.4
Bio8	Mean temperature of wettest quarter (°C)	5	12.5
Bio3	Isothermality (bio02/bio07) (×100) (°C)	3.8	3.5
UV-B	Annual mean UV-B (kJ/m^2^)	3.5	32.6
Elev	Elevation (m)	2.9	2.6
Bio10	Mean temperature of the warmest quarter (°C)	2.2	4.6
Bio15	Precipitation seasonality (coefficient of variation)	2.1	2.9
SpH	soil pH	1.3	1.1
Bio17	Precipitation of the driest quarter (mm)	1.3	6.5

**Table 3 plants-12-03999-t003:** List of environmental variables used in this research.

Type	Variable	Description (Unit)
Bioclimatic variable	Bio1	Annual mean temperature (°C)
	Bio2	Monthly diurnal range (mean of monthly (max temp − min temp)) (°C)
	Bio3	Isothermality (bio02/bio07) (×100) (°C)
	Bio4	Temperature seasonality (standard deviation × 100) (°C)
	Bio5	Max temperature of the warmest month (°C)
	Bio6	Min temperature of the coldest month (°C)
	Bio7	Temperature annual range (Bio5–Bio6) (°C)
	Bio8	Mean temperature of wettest quarter (°C)
	Bio9	Mean temperature of the driest quarter (°C)
	Bio10	Mean temperature of the warmest quarter (°C)
	Bio11	Mean temperature of the coldest quarter (°C)
	Bio12	Annual precipitation (mm)
	Bio13	Precipitation of the wettest month (mm)
	Bio14	Precipitation of the driest month (mm)
	Bio15	Precipitation seasonality (coefficient of variation)
	Bio16	Precipitation of the wettest quarter (mm)
	Bio17	Precipitation of the driest quarter (mm)
	Bio18	Precipitation of the warmest quarter (mm)
	Bio19	Precipitation of the coldest quarter (mm)
	Elev	Elevation (m)
Landform variable	SpH	soil pH
	SC	soil organic carbon (%)
Radiation variable	UV-B	Annual mean UV-B (kJ/m^2^)

## Data Availability

Data is contained within the article.

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
