# Peer review of "Predicting the Potential Distribution of Oxalis debilis Kunth, an Invasive Species in China with a Maximum Entropy Model"

_plants, 2023, doi:10.3390/plants12233999_

Round 1
Reviewer 1 Report
Comments and Suggestions for Authors
The topic of the manuscript is relevant. The amount of data is sufficient. However, it needs some revision to provide sufficient information to the reader.
The title corresponds to the presented findings.
Materials and methods are describe precisely research material and methods. Statistical methods are appropriate.
Results are presented correctly. However, in figures and tables, please always explain all abbreviations, even if they were explained in the text. Please make sure that a table or figure can be fully understood even without reading the text. In Figures, please name all axes. Figures must be clear without description – clear names of axis, detailed notes, etc.
Names of the location in the text tells nothing for the reader of other countries. Consideration should be given to present the locations indicating their main differences or similarities (soil, climate, edapho-climatic conditions).
Names of the location should remain only in the methodological part.
Author Response
|
Response to Reviewer X Comments
|
||
|
1. Summary |
|
|
|
Thank you very much for taking the time to review this manuscript. Please find the detailed responses below and the corresponding revisions.
|
||
|
2. Questions for General Evaluation |
Reviewer’s Evaluation |
Response and Revisions |
|
Does the introduction provide sufficient background and include all relevant references? |
Can be improved |
[In response to your suggestions, we have expanded the number and scope of references, while striving for a more seamless integration with this study.] |
|
Are all the cited references relevant to the research? |
Can be improved |
[In response to your suggestions, we have expanded the number and scope of references, while striving for a more seamless integration with this study.] |
|
Is the research design appropriate? |
Yes |
|
|
Are the methods adequately described? |
Yes |
|
|
Are the results clearly presented? |
Yes |
|
|
Are the conclusions supported by the results? |
Yes |
|
|
3. Point-by-point response to Comments and Suggestions for Authors |
||
|
Comments 1: [Results are presented correctly. However, in figures and tables, please always explain all abbreviations, even if they were explained in the text. Please make sure that a table or figure can be fully understood even without reading the text. In Figures, please name all axes. Figures must be clear without description – clear names of axis, detailed notes, etc.
Names of the location in the text tells nothing for the reader of other countries. Consideration should be given to present the locations indicating their main differences or similarities (soil, climate, edapho-climatic conditions).
Names of the location should remain only in the methodological part. ]
|
||
|
Response 1:Thank you for pointing this out. We agree with this comment. Therefore, we have modified the pictures and tables, adding clear axis names, detailed notes. [Figure 1. Page 2 : I have added explanation details. Figure 2. Page 3: I have increased the picture clarity and detail. Table 1. Page 5: I have clearly marked the coordinate axis as per your suggestion. Table 2. Page 5: I have enriched the details of the table. Figure 5. Page 7:I revised the horizontal and vertical coordinates to make the picture clear. Figure 7. Page 10: I modified the picture title.]” |
||
|
Comments 2: [Names of the location in the text tells nothing for the reader of other countries. Consideration should be given to present the locations indicating their main differences or similarities (soil, climate, edapho-climatic conditions).] |
||
|
Response 2: Agree. We have revised the part about place names, and only retained the most representative ones for reference. At the same time, we have made a simple analysis of their climate (temperature, soil) according to your suggestion. [Page 8, Lines 222-236: We have discussed and analyzed the climatic conditions of the suitable land.] |
||
|
Comments 3: [Names of the location should remain only in the methodological part.] |
||
|
Response 3: Agree. We have revised the part about place names. [Page 1, Lines 20: We have changed the place name. Page 1, Lines 38-39: We only keep a very representative reference of the famous works.] |
||
|
4. Response to Comments on the Quality of English Language |
||
|
Point 1: You have no relevant comments. |
||
|
Response 1: We have made a large amount of modifications and refinements to the article, and marked them in red. Please kindly check the attachment. |
||
|
5. Additional clarifications |
||
|
[Thank you so much for your valuable suggestions, we have made numerous adjustments based on your insightful feedback. Firstly, we have adjusted the order of the research structure. And we have expanded the number and scope of references, while striving for a more seamless integration with this study. Concurrently, we have augmented the content of both the introduction, results and discussion sections, intensified our analysis and interpretation of results, and provided recommendations for controlling Oxalis debilis Kunth. The details are provided in the attachment, please have a look. |
||

Reviewer 2 Report
Comments and Suggestions for Authors
Dear Authors
Reviewer report:
Regarding the manuscript entitled “Predicting the potential distribution of Oxalis corymbosa DC., an invasive species in China with a maximum entropy model” with ID plants-2071074
The present study aimed to study the distribution of the invasive plant Oxalis debilis in different regions of China and predict its future expansion using MaxEnt and ArcGIS. The topic is interesting as the invasive plant species become a hot topic for most researchers, scientists, and policymakers all over the world. The most serious issue is that authors use the Heterotypic Synonym of the Oxalis debilis. First and most importantly the accepted name for “Oxalis corymbosa DC.” is “Oxalis debilis Kunth” according to the Royal Botanic Graden (Plants of the World) https://powo.science.kew.org/. Therefore, the accepted name should be used instead of the Heterotypic Synonym. The introduction needs to be tightened with more information relevant to the manuscript topic. Results, figures, and tables are not presented well. The discussion section needs to be adjusted to focus on the main findings as well as supported with relevant citations. The paper lacks recommendations and suggestions to control or avoid more invasion by the Oxalis debilis. Therefore, I recommend a major revision of the article.
Sincerely Yours,
I proposed some suggestions that could improve the manuscript
Abstract:
· The Abstract doesn’t have significant data on the work and still does not present the paper. Please, rewrite with more details and in a tightened manner with the conclusion.
Keywords:
· The keywords should be corrected without repetition from the title.
Introduction:
· The introduction section needs to be tightened and provided with more information.
· The aim of the work is not clear. Please, clearly rewrite this, maybe as points.
· Figure 1: the legend needs to be revised correctly. Also, the scale is not unified and seems not correct as the leaves' size is different within the various pictures of the plant. What is meant by “On the lawn”? strange here!!. Fig. 1a: it is a whole plant with a shoot and root system. “In the flower groove”??? revise.
Materials and methods:
· Page 3, Lines 67-69: So primitive, please adjust this section in a proper manner.
· Figure 2: the figure resolution and color selection are not good. Please, use clear font and colors.
· Page 4, Lines 80 & 86: remove the symbol and use direct numbering.
· Page 4, Line 88: revise references in range.
· Table 1: no need for this table within the manuscript. It can be shifted to supplementary materials.
Results
· Figure 3: This figure resolution is not good. The scale is not clear. The small included map inside the figure should be clarified in the legend or changed to be self-representative.
· Figure 4: so primitive to be included in the manuscript.
· Figure 6: so primitive to be included in the manuscript.
· Table 2: what are the values within the table? Does it P value?
· The results are not presented well. This section needs major revisions and rewritten in a clear and concise manner.
Discussion
· Page 11, Line 275-24: all this section is not supported with relevant references.
· Page 12, Line 14: Please, revise.
· The whole section needs to be adjusted to focus on the main findings as well as supported with relevant citations.
Conclusion.
· I hope to see what are the recommendations and suggestions to control or avoid more invasion by the Oxalis debilis.
References
The References are formatted well, but some references have DOI and others do not.
Comments on the Quality of English Language
Moderate revision of the language is recommended
Author Response
|
Response to Reviewer X Comments
|
||
|
1. Summary |
|
|
|
Thank you very much for taking the time to review this manuscript. Please find the detailed responses below and the corresponding revisions.
|
||
|
2. Questions for General Evaluation |
Reviewer’s Evaluation |
Response and Revisions |
|
Does the introduction provide sufficient background and include all relevant references? |
Must be improved |
[In response to your suggestions, we have expanded the number and scope of references, while striving for a more seamless integration with this study] |
|
Are all the cited references relevant to the research? |
Can be improved |
In response to your suggestions, we have expanded the number and scope of references, while striving for a more seamless integration with this study |
|
Is the research design appropriate? |
Can be improved |
The experimental process has been improved. |
|
Are the methods adequately described? |
Can be improved |
A more detailed description of the experimental process was given. |
|
Are the results clearly presented? |
Must be improved |
We have made a significant improvement in the results. |
|
Are the conclusions supported by the results? |
Must be improved |
We have made a significant improvement in the results. |
|
3. Point-by-point response to Comments and Suggestions for Authors |
||
|
Comments 1: [The most serious issue is that authors use the Heterotypic Synonym of the Oxalis debilis.First and most importantly the accepted name for “Oxalis corymbosa DC.” is “Oxalis debilis Kunth” according to the Royal Botanic Graden] |
||
|
Response 1: [Replace "Oxalis corymbosa DC." with "Oxalis debilis Kunth"] Thank you for pointing this out. We have made a full-text revision on this issue. [Page 1, Lines 2] |
||
|
Comments 2: [ The Abstract doesn’t have significant data on the work and still does not present the paper. Please, rewrite with more details and in a tightened manner with the conclusion.] |
||
|
Response 2: Agree. The abstract has been revised to provide a more comprehensive presentation, encompassing crucial data and succinct conclusions of the study. [Page 1, Lines 8-27] |
||
|
Comments 3: [ The keywords should be corrected without repetition from the title.] |
||
|
Response 3: Your valuable suggestion is pretty good. We added ‘Suitable habitat’. But we think we should still retain certain keywords to facilitate retrieval. [Page 1, Lines 28-29] |
||
|
Comments 4: [ Introduction: · The introduction section needs to be tightened and provided with more information. · The aim of the work is not clear. Please, clearly rewrite this, maybe as points. · Figure 1: the legend needs to be revised correctly. Also, the scale is not unified and seems not correct as the leaves' size is different within the various pictures of the plant. What is meant by “On the lawn”? strange here!!. Fig. 1a: it is a whole plant with a shoot and root system. “In the flower groove”??? revise.] |
||
|
Response 4: Thank you for your valuable suggestions. We have made the following modifications according to your suggestions. [Page 1, Lines 32-43: References have been added. Page 2, Lines 51-65: We added the context and impact of Oxalis debilis Kunth as an invasive plant. Page 2,Figure 1.: I have added explanation details. Page 3, Lines 80-84: Modify the objective. ] |
||
|
Comments 5: [Materials and methods: · Page 3, Lines 67-69: So primitive, please adjust this section in a proper manner. · Figure 2: the figure resolution and color selection are not good. Please, use clear font and colors. · Page 4, Lines 80 & 86: remove the symbol and use direct numbering. · Page 4, Line 88: revise references in range. · Table 1: no need for this table within the manuscript. It can be shifted to supplementary materials.] |
||
|
Response 5: We agreed with them, and the modifications have been made as per your suggestions. Page 9, Change Figure 2 to Figure 6: Change font color Page 9, Lines 273-277: remove the symbol and use direct numbering. Page 10, Lines 279: The literature helps. Maybe we can keep it. Page 10, Change Table 1 to Table 3:Table 3 gives the reader a clearer idea of the selected data. May be retained ] |
||
|
Comments 6:[ Results · Figure 3: This figure resolution is not good. The scale is not clear. The small included map inside the figure should be clarified in the legend or changed to be self-representative. · Figure 4: so primitive to be included in the manuscript. · Figure 6: so primitive to be included in the manuscript. · Table 2: what are the values within the table? Does it P value? · The results are not presented well. This section needs major revisions and rewritten in a clear and concise manner.] |
||
|
Response 6: Agreed, We have made modifications according to your suggestions. |
||
|
Comments 7: [ Discussion · Page 11, Line 275-24: all this section is not supported with relevant references. · Page 12, Line 14: Please, revise. · The whole section needs to be adjusted to focus on the main findings as well as supported with relevant citations.] |
||
|
Response 7: The proposal you sent has been received and we wholeheartedly agree with it. Thanks for your suggestion, we have made a major revision to the discussion. |
||
|
Comments 8: [ Conclusion. · I hope to see what are the recommendations and suggestions to control or avoid more invasion by the Oxalis debilis.] |
||
|
Response 8: Agreed. Thank you for your valuable input, we put this part in ‘3.3.Comprehensive treatment of O. debilis’. |
||
|
Comments 9: [ References The References are formatted well, but some references have DOI and others do not.] |
||
|
Response 9: We totally agreed with you. We have tried our best and added many references as support, but due to the long history, some of the references do not have DOI. |
||
|
4. Response to Comments on the Quality of English Language |
||
|
Point 1: Moderate revision of the language is recommended |
||
|
Response 1: We have made a large amount of modifications and refinements to the article, and marked them in red. Please kindly check the attachment. |
||
|
5. Additional clarifications |
||
|
[Thank you so much for your valuable suggestions, we have made numerous adjustments based on your insightful feedback. Firstly, we have adjusted the order of the research structure. And we have expanded the number and scope of references, while striving for a more seamless integration with this study. Concurrently, we have augmented the content of all the introduction, results and discussion sections, intensified our analysis and interpretation of results, and provided recommendations for controlling Oxalis debilis Kunth. More details are provided in the attachment, please have a look.] |
||

Reviewer 3 Report
Comments and Suggestions for Authors
Authors, please consider the corrections noted in the manuscript. Also explain in detail some of the doubts expressed on the manuscript.
These types of studies are very common today, so they have lost their originality. However, this does not mean that they are not interesting, but even more important. Invasive species will continue to cause problems around the world, so it will be necessary to address them from this perspective. My only strong question is in relation to the spatial resolution, it seems to me that it should have been done at 1 km2 and not at 25 km2. Thus, the distribution of the species is far from being close to the current and future distributions.
Author Response
|
Response to Reviewer X Comments
|
||
|
1. Summary |
|
|
|
Thank you very much for taking the time to review this manuscript. Please find the detailed responses below and the corresponding revisions.
|
||
|
2. Questions for General Evaluation |
Reviewer’s Evaluation |
Response and Revisions |
|
Does the introduction provide sufficient background and include all relevant references? |
Yes |
|
|
Are all the cited references relevant to the research? |
Yes |
|
|
Is the research design appropriate? |
Yes |
|
|
Are the methods adequately described? |
Yes |
|
|
Are the results clearly presented? |
Yes |
|
|
Are the conclusions supported by the results? |
Yes |
|
|
3. Point-by-point response to Comments and Suggestions for Authors |
||
|
Comments 1: [My only strong question is in relation to the spatial resolution, it seems to me that it should have been done at 1 km2 and not at 25 km2. Thus, the distribution of the species is far from being close to the current and future distributions.] |
||
|
Response 1: [Replace "25 km2" with "every 2.5 arc-minutes"] Thank you for pointing this out. We have unified spatial resolution. |
||
|
4. Response to Comments on the Quality of English Language |
||
|
Point 1: Moderate revision of the language is recommended |
||
|
Response 1: We have made a large amount of modifications and refinements to the article, and marked them in red. Please kindly check the attachment. |
||
|
5. Additional clarifications |
||
|
[Thank you so much for your valuable suggestions, we have made numerous adjustments based on your insightful feedback. Firstly, we have adjusted the order of the research structure. And we have expanded the number and scope of references, while striving for a more seamless integration with this study. Concurrently, we have augmented the content of both the introduction, results and discussion sections, intensified our analysis and interpretation of results, and provided recommendations for controlling Oxalis debilis Kunth. The details are provided in the attachment, please have a look.] |
||

Reviewer 4 Report
Comments and Suggestions for Authors
Your article presents a study on the potential distribution of Oxalis corimbosa DC. (= O. debilis Kunth) through maximum entropy modelling.
You applied well known methods to collect and process occurrence data, however the text is flawed by a number of issues.
To begin with, little consideration is given to the study species. The scientific name used throughout the text is currently regarded as synonym of Oxalis debilis Kunth and the species' native range is Central and South America, but you provide no such information.
Then, you use MaxEnt to model the predicted potential distribution of the species under the current environmental conditions, then use the prediction to make inferences as to the areas subjected to potential threat by the expansion of the species. For such a goal, an ENM (Broenniman et al, 2012) might be a better choice.
Materials and Methods are a bit unclear and sometimes inconsistent: e.g. in Fig. 2 the spatial resolution of the environmental variables is 2.5 m, while in the text it is 2.5 arc-minutes (misspelled as are-minutes) or approximately 4.5 km, then you mention a county level and finally a buffer of 3 km. You mention the reduction of spatial correlation, but you do not provide any metric for it.
You do not provide a method for the calculation of contribution rate and in section 2.2.3 wrongly associate Maxent output value with model performance.
Your results contain an extensive repetition of what you already presented in previous sections and some confusion between potential distribution range and areas suitable for species growth. Also the discussion incudes a repetition of other sections and is surprisingly nearly devoid of references (only two are mentioned).
English language is generally adequate and the references might be improved.
Please see the annotated text (enclosed) for further details and comments.
In light of its issues, I am afraid that the article cannot be published on Plants.

Author Response
|
Response to Reviewer X Comments
|
||
|
1. Summary |
|
|
|
Thank you very much for taking the time to review this manuscript. Please find the detailed responses below and the corresponding revisions.
|
||
|
2. Questions for General Evaluation |
Reviewer’s Evaluation |
Response and Revisions |
|
Does the introduction provide sufficient background and include all relevant references? |
Can be improved |
[In response to your suggestions, we have expanded the number and scope of references, while striving for a more seamless integration with this study] |
|
Are all the cited references relevant to the research? |
Must be improved |
In response to your suggestions, we have expanded the number and scope of references, while striving for a more seamless integration with this study |
|
Is the research design appropriate? |
Can be improved |
The experimental process has been improved. |
|
Are the methods adequately described? |
Must be improved |
A more detailed description of the experimental process was given. |
|
Are the results clearly presented? |
Must be improved |
We have made a significant improvement in the results. |
|
Are the conclusions supported by the results? |
Must be improved |
We have made a significant improvement in the results. |
|
3. Point-by-point response to Comments and Suggestions for Authors |
||
|
Comments 1: [To begin with, little consideration is given to the study species. The scientific name used throughout the text is currently regarded as synonym of Oxalis debilis Kunth and the species' native range is Central and South America, but you provide no such information.] |
||
|
Response 1: [Replace "Oxalis corymbosa DC." with "Oxalis debilis Kunth". Add introductions to the species' native distribution and relevant references.] Thank you for pointing this out. We have made a full-text revision on this issue. [Page 1, Lines 2 :Replace "Oxalis corymbosa DC." with "Oxalis debilis Kunth", and modified the full text. Page 1, Lines 32-37: Add introductions to the species' native distribution and relevant references.] |
||
|
Comments 2: [ Then, you use MaxEnt to model the predicted potential distribution of the species under the current environmental conditions, then use the prediction to make inferences as to the areas subjected to potential threat by the expansion of the species. For such a goal, an ENM (Broenniman et al, 2012) might be a better choice.] |
||
|
Response 2: We agree with you, but these recently published papers on distribution prediction using the MaxEnt model can prove that the model is reliable. [Ling et al, 2023. Predicting Ecologically Suitable Areas of Cotton Cultivation Using the MaxEnt Model in Xinjiang, China. Ecologies, 4(4),654-670 https://doi.org/10.3390/ecologies4040043 Shi et al, 2023. Prediction of the potentially suitable areas of Litsea cubeba in China based on future climate change using the optimized MaxEnt model. Ecological Indicators, 148. https://doi.org/10.1016/j.ecolind.2023.110093 Butoto and Jun Won Kang, 2023. MaxEnt modeling for predicting the potential distribution of Lebrunia bushaie Staner (Clusiaceae) under different climate change scenarios in Democratic Republic of Congo. Asia-Pacific Biodiversity, https://doi.org/10.1016/j.japb.2023.06.005] |
||
|
Comments 3: [ Materials and Methods are a bit unclear and sometimes inconsistent: e.g. in Fig. 2 the spatial resolution of the environmental variables is 2.5 m, while in the text it is 2.5 arc-minutes (misspelled as are-minutes) or approximately 4.5 km, then you mention a county level and finally a buffer of 3 km. You mention the reduction of spatial correlation, but you do not provide any metric for it.] |
||
|
Response 3: [We unify the spatial resolution to ‘2.5 arc-minutes’.] Thank you for pointing this out. We have unified spatial resolution. [Page 10, Lines 295, 306: Unified spatial resolution. Delete the description ‘approximately 4.5 km, then you mention a county level and finally a buffer of 3 km‘.] |
||
|
Comments 4: [You do not provide a method for the calculation of contribution rate and in section 2.2.3 wrongly associate Maxent output value with model performance.] |
||
|
Response 4: Thank you for your valuable suggestions. Contribution rate automatically calculated by MaxEnt model. 2.2.3 has been modified. Please see the attachment for details. [Page 12, Lines 352-362: The original 2.3.3 is divided into two parts: ‘Maxent model accuracy checking’ and ‘Classification of the suitable habitat ranks’, which is more logically clear. ] |
||
|
Comments 5: [Your results contain an extensive repetition of what you already presented in previous sections and some confusion between potential distribution range and areas suitable for species growth. Also the discussion incudes a repetition of other sections and is surprisingly nearly devoid of references (only two are mentioned).] |
||
|
Response 5: Thank you very much for your suggestion. We reclassified and graded the potential distribution (suitable habitat) calculated by our model for better observation and discussion. Misleading words have been changed and replaced. Meanwhile, the discussion has been greatly modified, divided into three parts, and cited relevant references. Page 7-8, Lines199-260: Discussion is divided into three parts, and has been greatly modified] |
||
|
4. Response to Comments on the Quality of English Language |
||
|
Point 1: English language is generally adequate. |
||
|
Response 1: We have made a large amount of modifications and refinements to the article, and marked them in red. Please kindly check the attachment. |
||
|
5. Additional clarifications |
||
|
[Thank you so much for your valuable suggestions, we have made numerous adjustments based on your insightful feedback. Firstly, we have adjusted the order of the research structure. And we have expanded the number and scope of references, while striving for a more seamless integration with this study. Concurrently, we have augmented the content of all the introduction, results and discussion sections, intensified our analysis and interpretation of results, and provided recommendations for controlling Oxalis debilis Kunth. The details are provided in the attachment, please have a look.] |
||

Round 2
Reviewer 2 Report
Comments and Suggestions for Authors
Dear Ms. Supakorn Nundaeng
Assistent Editor
Plants, MDPI
Reviewer report:
Regarding the manuscript entitled “Predicting the potential distribution of Oxalis corymbosa DC., an invasive species in China with a maximum entropy model” with ID plants-2071074
Most of the requested corrections and comments were concerned, however, still some comments were not revised and corrected such as The keywords should be corrected without repetition from the title. Otherwise, the paper is redady for publication.
Comments on the Quality of English LanguageEnglish is good
Author Response
|
1. Summary |
|
|
|
Thank you so much for taking the time out of your busy schedule to review this manuscript. Please find the detailed responses below and the corresponding revisions.
|
||
|
2. Questions for General Evaluation |
Reviewer’s Evaluation |
Response and Revisions |
|
Does the introduction provide sufficient background and include all relevant references? |
Yes |
|
|
Are all the cited references relevant to the research? |
Yes |
|
|
Is the research design appropriate? |
Yes |
|
|
Are the methods adequately described? |
Yes |
|
|
Are the results clearly presented? |
Yes |
|
|
Are the conclusions supported by the results? |
Yes |
|
|
3. Point-by-point response to Comments and Suggestions for Authors |
||
|
Comments 1: [The keywords should be corrected without repetition from the title] |
||
|
Response 1: [Replace keywords with " Potential geographic area; MaxEnt model; Environmental variable; invasive plant; Alien species; Suitable habitat"] Thank you for pointing this out. The key words have been revised to avoid repetition with the title. [Page 1, Lines 28-29] |
||
|
4. Response to Comments on the Quality of English Language |
||
|
Point 1: Moderate revision of the language is recommended |
||
|
Response 1: We have made modifications and refinements to the article. |
||
|
5. Additional clarifications |
||
|
[Thank you so much for your valuable suggestions, we have made numerous adjustments based on your insightful feedback. The keywords have been modified based on your suggestion, ensuring that they do not overlap with the title. This revision focuses on modifying certain sentences and words. The last revision made adjustments to the order of the research structure, expanded both the number and scope of references, increased the content in the introduction, results, and discussion sections, enhanced the analysis and interpretation of results, and provided suggestions for controlling Oxalis debilis Kunth. The attachment provides more detailed information. Thank you for taking the time to check it amidst your busy schedule. Please kindly review it.] |
||

Reviewer 4 Report
Comments and Suggestions for Authors
I congratulate you with the improved text of your paper. Few issues remain to be solved, as pointed out in the annotated typescript here enclosed. The paper can be published after the mentioned issues have been duly addressed.

Please correct wrong sentences and improve clarity according to the suggestions in the annotated typescript.
Author Response
|
1. Summary |
|
|
|
Thank you so much for taking the time out of your busy schedule to review this manuscript. Please find the detailed responses below and the corresponding revisions.
|
||
|
2. Questions for General Evaluation |
Reviewer’s Evaluation |
Response and Revisions |
|
Does the introduction provide sufficient background and include all relevant references? |
Yes |
|
|
Are all the cited references relevant to the research? |
Yes |
|
|
Is the research design appropriate? |
Yes |
|
|
Are the methods adequately described? |
Yes |
|
|
Are the results clearly presented? |
Can be improved |
The modification has been made based on your suggestions. Please refer to the attachment for further details. |
|
Are the conclusions supported by the results? |
Yes |
|
|
3. Point-by-point response to Comments and Suggestions for Authors |
||
|
Comments 1: [The format of "Oxalis debilis Kunth" needs to be revised.] |
||
|
Response 1: Thank you for pointing this out. The "Oxalis debilis" is written in italics. [Page 4, Lines 120; Page 6, Lines 155] |
||
|
Comments 2: [Bio14 contribution is a full order of magnitude higher than the others. You should point out this finding and possibly add a paragraph to discussion to comment on it.] |
||
|
Response 2: Thank you for your valuable advice. We have emphasized in the Results and analyzed in the Discussion. [Page 5, Lines 142: The difference of Bio14 was emphasized. Page 8, Lines 229-234: The analysis of bio14 has been conducted.] |
||
|
4. Response to Comments on the Quality of English Language |
||
|
Point 1: Please correct wrong sentences and improve clarity according to the suggestions in the annotated typescript. |
||
|
Response 1: The suggestions you provided have been extremely helpful to us. We have made the necessary modifications based on your valuable input. |
||
|
5. Additional clarifications |
||
|
[Thank you so much for your invaluable suggestions. We have made specific adjustments based on your valuable opinions. The keywords have been modified based on your suggestion, ensuring that they do not overlap with the title. This revision focuses on modifying specific sentences and words. The last revision made adjustments to the order of the research structure, expanded both the number and scope of references, increased the content in the introduction, results, and discussion sections, enhanced the analysis and interpretation of results, and provided suggestions for controlling Oxalis debilis Kunth.
The attachment provides more detailed information. The advice you gave me has been incredibly helpful. Thank you for taking the time to check it despite your busy schedule. Could you please kindly review it?] |
||
